# Energy Partitioning and Latent Heat Flux Driving Factors of the CAM Plant Pineapple (*Ananas comosus* (L.) Merril) Grown in the South Subtropical China

**DOI:** 10.3390/plants13010021

**Published:** 2023-12-20

**Authors:** Zhigang Liu, Baoshan Zhao, Haofang Yan, Junbo Su

**Affiliations:** 1South Subtropical Crops Research Institute, Chinese Academy of Tropical Agricultural Sciences, Zhanjiang 524091, China; liuzhigangak47@163.com (Z.L.); zhao_baoshan@outlook.com (B.Z.); 2Long’an Yangpu Agricultural Technology Co., Ltd., Nanning 532704, China; 3Zhanjiang Experimental Station, Chinese Academy of Tropical Agricultural Sciences, Guangdong Engineering Technology Research Center for Dryland and Water Saving Agriculture, Zhanjiang Experimental and Observation Station for National Long-Term Agricultural Green Development, Zhanjiang 524091, China; 4Research Center of Fluid Machinery Engineering and Technology, Jiangsu University, Zhenjiang 212013, China; 1000004265@ujs.edu.cn

**Keywords:** energy partitioning, evapotranspiration, CAM plant, tropical fruits, Bowen ratio, environmental factors

## Abstract

Elucidation of different vegetation energy partitioning and environmental control factors at the agro-ecosystem levels is critical for better understanding and scientific management of farmland. Pineapple (*Ananas comosus* (L.) Merril) is a tropical plant widely cultivated in the southern subtropical region of China; however, the energy partitioning of crassulacean acid metabolism (CAM) plants like pineapple and their interactions with the environment remain not well understood. In this study, we investigated the energy partitioning patterns of pineapple fields and latent heat flux (*LET*) response to environmental factors using the Bowen ratio energy balance system and meteorological observation field data. The results showed that the CAM plant pineapple energy partitioning was significantly different from the common C_3_ and C_4_ crops during the study period, which was mainly attributed to the complex interactions between CAM plant transpiration and the environment. Specifically, sensible heat flux was the main component of net radiation (*R_n_*), followed by the *LET*, accounting for 65.0% and 30.8% of the *R_n_*, respectively. Soil heat flux accounts for a very small fraction (4.2%). The mean values of the Bowen ratio were 2.09 and 1.41 for sunny and cloudy days during the daytime and 0.74 and 0.46 at night, respectively. *LET* is a key factor in responding to crop growth status and agricultural water management, and the path analysis indicates that its variation is mainly influenced directly by *R_n_* with a direct path coefficient of 0.94 on sunny days, followed by vapor pressure deficit, air temperature and relative humidity, which indirectly affect *LET* through the *R_n_* pathway, whereas soil moisture and wind speed have a low effect on *LET*. On cloudy days, the effect of *R_n_* on *LET* was overwhelmingly dominant, with a direct path coefficient of 0.91. The direct path coefficients of the remaining factors on *LET* were very small and negative. Overall, this study is an important reference for enhancing the impact of pineapple as well as CAM plants on the surface energy balance and regional climate.

## 1. Introduction

Land surface energy partitioning plays a pivotal role in surface–atmosphere interactions [1,2], exerting control over terrestrial ecosystem temperature, water transport, vegetation growth and ecosystem productivity. It also exerts influence over surface atmospheric circulation [3,4,5]. As an important input parameter for regional climate models and ecohydrological models [6], it is critically important to understand the energy partitioning mechanisms in various ecosystems with diverse surface conditions.

Agro-ecosystems, being fundamental to human social development, constitute a significant component of global ecology, energy balance and regional climate studies [7]. Energy fluxes within agro-ecosystems maintain a delicate equilibrium between matter and energy. The processes of water and heat transfer within these systems exert a profound impact on crop growth and the formation of crop yields [8,9]. Given the intricate interplay between energy fluxes, crops and environmental variables (such as temperature, precipitation and solar radiation), an objective assessment of surface–atmosphere energy partitioning assumes paramount importance in the realms of agricultural water resource management and crop-environmental modeling [9]. Furthermore, it contributes substantially to the scientific understanding of agro-ecosystem functioning [10,11]. In recent years, many studies on energy fluxes in agro-ecosystems have focused on the mechanisms of water–heat exchange [12,13,14], changes in carbon stocks in crops and soils [15,16,17], differentiation of flux contributions in agro-ecosystems and influencing factors [18,19,20]. These studies rely heavily on simplified energy balance equations where net radiation (*R_n_*) energy from the surface received is consumed through sensible heat flux (*H*), latent heat flux (*LET*) and soil heat flux (*G*).

Accurate monitoring of surface energy fluxes is fundamental to the study of energy partitioning and its response mechanisms. Energy fluxes can be measured by a variety of techniques, such as eddy correlation and the Bowen ratio energy balance (BREB) system [21]. The BREB method has emerged as a standard tool for continuously and conveniently measuring energy fluxes across large, homogenized surface areas. This technique has found successful applications in the study of energy fluxes, their distribution, and water–heat exchange in both grassland and agricultural ecosystems [22,23,24,25]. The Bowen ratio (*β* = *H*/*LET*) is a crucial parameter of energy partitioning that reflects the ratio of sensible and latent heat distribution [21]. *LET*, which links water consumption in agro-ecosystems to the energy driving evapotranspiration [23], holds critical importance for agricultural water management, crop growth and yield [24]. Understanding and distinguishing the influencing factors affecting *LET* in agricultural fields constitutes a significant scientific challenge in the realm of agricultural water management.

Path analysis emerges as an efficient method for examining interactions between independent variables and multiple dependent variables [26,27]. It integrates statistical techniques from factor analysis and linear regression analysis to identify and validate causal models. This approach allows for the categorization of correlation coefficients between causal variables into direct effects (direct path coefficients) and indirect effects (indirect path coefficients). Through pathway analysis, if the direct path coefficient of a factor is greater than the sum of the indirect path coefficients, it means that the effect of this factor on the dependent variable is reflected in a direct effect, while the opposite is an indirect effect. Such categorization enables the analysis of the direct and indirect significance of independent variables on the dependent variable [28,29]. Path analysis overcomes the limitations inherent in commonly used methods like multiple linear regression analysis and correlation analysis, which fail to quantitatively distinguish direct and indirect influencing factors and paths affecting *LET* [30]. In this study, we propose to use correlation analysis in conjunction with path analysis to better quantify the impact of environmental factors on *LET*.

Pineapple (*Ananas comosus* (L.) Merril) ranks as the third-largest tropical fruit in global production, following banana and mango, and holds a dominant position in the trade of tropical fruits [31]. The southern subtropical region of China is located in the transitional zone between subtropical and tropical climates. This region is characterized by high temperatures, elevated humidity levels and uneven spatial and temporal distribution of precipitation [32]. With the increase in market demand in recent years, the planting area and yield of pineapples in this region have shown an upward trend, with approximately 40 × 10^3^ hectares planted in 2022. Pineapple stands out as a prime example of a crassulacean acid metabolism (CAM) crop. During daylight hours, when solar radiation and temperatures peak, the leaf stomata of pineapples close to reduce transpiration and increase water use efficiency. Conversely, at night, they open to facilitate carbon assimilation while minimizing transpiration [33]. These characteristics suggest that pineapples may exhibit unique patterns of transpiration and energy fluxes, potentially differing significantly from C_3_ and C_4_ crops. Previous studies on energy partitioning in CAM plants of the *Cactaceae* family, situated in semiarid environments, have reported higher *H* compared to *LET*. This contrasts with the common C_3_ and C_4_ crop energy partitioning, which is predominantly dominated by *LET* [34]. Previous studies on energy partitioning in agro-ecosystems have mainly focused on grain crops such as wheat, rice and maize under changing environments, and for pineapple, studies have been conducted to analyze gas exchange and stomatal conductance in leaf and plant physiology [35,36,37]. However, no studies have been conducted on the field-scale energy partitioning characteristics of pineapple in the southern subtropical environment. Therefore, this study aims to address this research gap by determining the energy partitioning patterns and the drivers of *LET* in pineapple fields in the southern subtropical region of China.

The primary objectives of this study were as follows: (1) to investigate the energy partitioning characteristics of pineapple fields in southern subtropical China using the BREB system and (2) to examine the factors influencing *LET* during the pineapple growing period. The findings of this research will contribute to an improved understanding of energy balance and transformation patterns in agro-ecosystems. Additionally, it serves as a basis for further assessments of the impact of CAM plants on surface energy balance and regional climate.

## 2. Materials and Methods

### 2.1. Study Site

The study site is located in Zhanjiang, the largest pineapple production area in China (Figure 1a). This region experiences a subtropical monsoon climate characterized by an average annual temperature of 23.6 °C, an average annual rainfall of 1619.6 mm, an average annual *ET*_0_ of 1242.7 mm, and an average annual sunshine duration of 1946.5 h. The majority of precipitation occurs from May to October, accounting for 83.1% of the annual precipitation [38].

The experiment was conducted from January 2022 to June 2023 at the Pineapple Experimental Base of the Institute of South Subtropical Crops Research Institute, Chinese Academy of Tropical Agricultural Sciences (21°09′ N, 110°16′ E). The experimental pineapple fields were located in the southwest of the pineapple base (Figure 1b,c), with an area of 7200 m^2^. The selected pineapple variety was the golden pineapple, a local main variety, transplanted in March 2022, with row spacing of 1.2 m × 0.5 m. The soil texture of the test area was classified as brick red loam, with an average soil dry bulk density of 1.23 g/cm^3^ at 0–90 cm depth and a field capacity of 25.37%. According to the growth of pineapple in the field, its reproductive period was divided into three phases: nutritive growth period (1 April 2022–31 December 2022), flowering and reddish period (1 January 2023–10 February 2023), and ripening period (11 February 2023–30 June 2023).

### 2.2. Field Measurements

Routine meteorological data were collected throughout the pineapple growing period using a meteorological observatory (Figure 1d). These observations encompassed standard meteorological parameters, including air temperature, relative humidity, radiation, wind speed, wind direction and precipitation.

The BREB system was used to observe the energy flux data of pineapple fields every 30 min. The study area is prevailing southeasterly winds, and the BREB system was arranged in the southwestern part of the test field to ensure a large fetch. *R_n_* at 2.5 m above ground level was observed by a net radiometer (NR Lite2, Kipp & Zonen, Delft, The Netherlands), and *G* was observed by a soil heat flux panel (HFP01, Hukseflux, Delft, The Netherlands) placed about 2 cm below the ground surface. To observe temperature and vapor pressure gradients, *T_a_* and *RH* at 1.5 m and 2.5 m were observed by a temperature and humidity sensor (HMP155A, Vaisala, Helsinki, Finland), wind speed and direction were measured by an anemometer (03002, RM Young, Traverse City, MI, USA) at 2.5 m, and *SWC* was measured by a soil moisture sensor (CS655, Campbell, Logan, UT, USA) placed at three different depths (5, 20, 50 cm) below the ground surface. All sensors were evaluated and calibrated prior to fielding, and to accurately determine air temperature and water pressure gradients, two temperature and humidity sensors were placed at the same height for calibration prior to the start of the experiment. All data were sampled every 1 min averaged over 30 min and recorded by a data logger (CR3000, Campbell, USA).

### 2.3. Data Processing

The BREB method was used to calculate pineapple field *H* and *LET* [39,40]. Firstly, in agro-ecosystems, the energy balance equation can be simplified as:(1)$$R_{n}=H+LET+G$$
where *R_n_* is the net radiation, W·m^−2^; *G* is the soil heat flux, W·m^−2^; *H* is the sensible heat flux, W·m^−2^; and *LET* is the latent heat flux, W·m^−2^.

Then, the Bowen ratio *β* is defined as the ratio of *H* and *LET*:(2)$$\beta =\frac{H}{LET}$$

*β* can be calculated by the following equation:(3)$$\beta =\gamma \frac{\partial T/\partial z}{\partial e/\partial z}=\gamma \frac{\Delta T}{\Delta e}$$
where *γ* is the psychrometric constant, kPa·°C^−1^, ∂T/∂z and ∂e/∂z are the temperature gradient and vapor pressure gradient, respectively. Δ*T* and Δ*e* are the temperature and vapor pressure differences, respectively, between the two measurement levels. 

Substitute Equation (2) into Equation (1), and deduce that *H* and *LET* are calculated using the following equation:(4)$$H=\frac{\beta }{1+\beta }(R_{n}-G)$$
(5)$$LET=\frac{1}{1+\beta }(R_{n}-G)$$

The BREB observation data were subject to validity screening based on what was previously reported by Perez et al. (Table 1) [41], and outliers of *H* and *LET* that fell outside the range of −50–700 W/m^2^ were rejected. The data exhibited a higher incidence of outliers before and after sunrise and sunset. The screened and rejected data were subsequently interpolated using linear interpolation to ensure completeness.

The data were organized and calculated using Excel 2019, charted using Origin 2021 software, and Pearson correlation coefficient calculation and path analysis were performed using IBM SPSS Statistics 26 software.

## 3. Results

### 3.1. Meteorological and Soil Moisture Conditions

Seasonal variations of meteorological factors and soil moisture levels during the pineapple growth period in 2022–2023 are shown in Figure 2. Net radiation (*R_n_*), air temperature (*T_a_*) and vapor pressure deficit (*VPD*) exhibited a similar temporal pattern, gradually increasing from January to a peak in July with mean values of 29.41 °C, 119.71 W·m^−2^ and 0.78 kPa, respectively, before gradually decreasing (Figure 2a,b,d). Relative humidity (*RH*) showed a monthly variation range of 68.7% to 90.7%, with its lowest point occurring in December (Figure 2c). Wind speed (*WS*) ranged from 0.31 to 1.94 m·s^−1^ without an obvious temporal trend, with a mean value of 1.19 m·s^−1^, and prevailing wind directions were southeast (Figure 2e). Precipitation was 775.44 mm in 2022, mainly occurring from April to June, and 545.4 mm from January to June 2023. Soil water content (*SWC*) exhibited a sudden increase following precipitation events, followed by a gradual decrease. The average soil water content at 5 cm, 10 cm and 20 cm depths was measured as 0.25, 0.25 and 0.30 m^3^·m^3^, respectively (Figure 2f).

### 3.2. Daily Variation of Energy Partitioning

The daily variation curves of energy fluxes during the vegetative, flowering and yield formation stages of pineapple on typical sunny days are shown in Figure 3. It can be seen that all components exhibit positive values during daylight hours and transition to negative values at night. The direction and magnitude of each energy transfer undergo changes before and after sunrise and sunset, with the intersection of positive and negative transitions aligning closely with daytime transitions. The energy in the pineapple fields in the study area was mainly consumed by sensible heat flux (*H*), and there was no significant change in energy distribution at different growth stages, which were *H* > *LET* > *G*.

*R_n_* is the primary source of energy income for pineapple fields, and its daily variation curve rises rapidly in the morning, reaching a peak around 12:00, then begins to decline and remains basically at −20~0 W·m^−2^ after 19:00 until 7:00 the following day. The ground releases energy to the atmosphere through longwave radiation, with a maximum of 786.50 W·m^−2^ during the daytime, and peaks at the vegetative and yield formation stages are higher than those observed during the flowering stage.

Daily variations in energy expenditure terms (*H*, *LET* and *G*) were smaller in magnitude than that of *R_n_*. *LET* increased in tandem with rising *R_n_* during daytime, peaking around 12:00–13:00, slightly later than the net radiation peak. Subsequently, *LET* gradually decreased, and it assumed negative values during nighttime. The variation range of *G* was limited and remained close to 0, with the peak occurring at 12:00–14:00. *G* lagged behind the time of peak *R_n_* by approximately 1 to 2 h. This delay is attributed to the larger specific heat capacity of the soil compared to the air, requiring additional time for energy transfer after absorbing net radiation.

The daily variation curves of energy fluxes at different growth stages on typical cloudy days are shown in Figure 4. It can be seen that due to the increasing influence of cloud cover on radiation, as the cloud cover fluctuates, the flux curves exhibit multi-peak characteristics during the day, and all flux values are significantly lower.

On cloudy days, energy income *R_n_* was significantly lower than on sunny days, with a daily average value of 34.60 W·m^−2^. The diurnal variation curves of *H*, *LET* and *G* closely followed the pattern of *R_n_*. It can be seen that *LET* exceeded *H* as the dominant energy expenditure when the daily maximum *R_n_* was less than 100 W·m^−2^, indicating that the heat exchange between the pineapple fields and the atmosphere is lower on cloudy days. During cloudy days, the maximum value of *G* reached approximately 10 W·m^−2^, typically peaking around 15:00 h–18:00 h, while the average value of *G* was negative at about −8 W·m^−2^. This observation indicates that in the environment of pineapple fields, energy income *R_n_* is low, and the soil releases heat to the atmosphere in order to make up for energy expenditures. This phenomenon effectively explains the occurrence of *LET* greater than *R_n_* on cloudy days. During cloudy mornings with high *R_n_*, energy is stored in the ground, and the soil continues to release energy to the atmosphere in the afternoon. This resulted in negative values for *G,* which served to supplement the lack of energy supplied for evapotranspiration, resulting in *LET* surpassing *R_n_*.

### 3.3. Energy Partitioning of Pineapple Fields during Different Growth Periods

The energy partitioning of pineapple fields during the vegetative, flowering and yield formation stages is shown in Figure 5 and Table 2. It can be seen that the *R_n_* absorbed during the vegetative and yield formation stages on sunny days is close to 413.44 W·m^−2^ and 431.33 W·m^−2^, respectively. In contrast, the flowering stage experienced the lowest period net radiation levels of the year, with an average daily net radiation of 227.36 W·m^−2^. Expenditures of *H* were the largest, followed by *LET*, while *G* remained relatively small in comparison to the net radiation absorbed during the day at all stages. This observation suggests that the transpiration capacity of the pineapple canopy was notably limited, and the net radiation absorbed in pineapple fields was mainly used for surface–atmosphere heat exchange. The average net radiation absorbed during the day was 357.38 W·m^−2^, in which *H* and *LET* accounted for 65.0% and 30.8%, respectively, and *G* accounted for only 4.2%. During nighttime, flux values were diminished, and their direction changed. The average net radiation was −35.41 W·m^−2^, with *LET* comprising the largest proportion at an average of −21.30 W·m^−2^, followed by *G* with an average of −8.56 W·m^−2^, and *H* registering as the smallest component with an average value of −5.43 W·m^−2^.

The *R_n_* absorbed by the pineapple fields was significantly lower on cloudy days, with a mean value of 76.84 W·m^−2^. Although *H* still dominated the energy expenditure during the daytime, the proportion of *LET* in the energy expenditure increased compared to sunny days, with the *H* and *LET* proportions of 58.3% and 44.1%, respectively. On cloudy days, the *G* was negative on average, and the soil released energy to the atmosphere.

### 3.4. Diurnal Variation of Bowen Ratio

The Bowen ratio (*β*) serves as an indicator of the energy partitioning ratio between *H* and *LET*. Figure 6 shows the daily variation curves of the *β* in pineapple fields under different weather conditions. It becomes evident that the trend of Bowen ratio changes closely mirrors the diurnal variations in energy fluxes. The *β* remains relatively low and stable during nighttime, gradually rising around 8:00 h and then decreasing around 18:00 h. During the sunrise and sunset periods, there are large fluctuations in *β*, with abnormal fluctuations in *LET* and *H*. This phenomenon seems more pronounced on sunny days and may cause significant errors in energy flux measurements. The *β* typically assumes positive values, indicating that latent and sensible heat fluxes generally occur in the same direction. During the daytime hours (8:00 h–18:00 h), the average *β* values were 2.09 and 1.41 for sunny and cloudy days, respectively. This suggests that the *R_n_* reaching the pineapple fields during the daytime primarily contributes to *H*. Conversely, during the nighttime hours (18:00 h–8:00 h), the average *β* values were 0.74 and 0.46 for sunny and cloudy days, respectively. During these nighttime periods, the pineapple canopy’s exchange of heat and the atmosphere is relatively lower, and energy partitioning is predominantly dominated by *LET*, which is consistent with the response of the daily variation curves of energy fluxes. Moreover, pineapple fields exhibit lower heat exchange with the atmosphere on cloudy days and during nighttime, resulting in smaller fluctuations in the *β*.

### 3.5. Driving Factors for LET

*LET* represents the energy form associated with evapotranspiration in agro-ecosystems, holding significant importance for agricultural water management and crop water use efficiency. Figure 7 shows the results of the correlation and path analysis of *LET* with corresponding environmental factors *R_n_*, *T_a_*, *RH*, *VPD*, *WS* and *SWC* for pineapple fields under both sunny and cloudy days.

On sunny days (Figure 7a), the absolute values of the correlation coefficients between *LET* and various environmental factors followed the order: *R_n_* (0.96) > *VPD* (0.61) > *T_a_* (0.36) > *WS* (0.34) > *RH* (0.32) > *SWC* (0.008). This indicates that *R_n_* was the most important environmental factor affecting *LET*, followed by *VPD*, *T_a_*, *WS* and *RH*. Although the effect of *SWC* on *LET* was statistically significant, the correlation remained low, likely due to the nonlinear nature of *SWC* changes on *LET*. Conversely, on cloudy days (Figure 7c), the absolute values of the correlation coefficients between various environmental factors and *LET* ranked as follows: *R_n_* (0.88) > *T_a_* (0.14) > *RH* (0.08) > *WS* (0.04) > *SWC* (0.035) > *VPD* (0.025). Notably, only *R_n_* exhibited a relatively strong correlation with *LET* on cloudy days, while the rest of the factors displayed weaker correlations. Except for the significant negative correlation between *RH* and *LET* on sunny days, *R_n_*, *T_a_*, *VPD* and *WS* exhibited positive correlations with *LET*.

On sunny days, path analysis results indicate that the direct path coefficients of *R_n_*, *T_a_*, *RH* and *VPD* on *LET* were positive (Figure 7b). *R_n_* has the greatest direct impact on *LET*, with a direct path coefficient of 0.94. The indirect path coefficient was significantly smaller than the direct path coefficient. In contrast, the direct path coefficients of the other factors in relation to *LET* were smaller and less than the sum of the indirect path coefficients. This implies that the effect of *R_n_* on *LET* is mainly attributed to its direct effect, while factors such as *VPD* were reflected in the indirect effect. On cloudy days, the effect of *R_n_* on *LET* was overwhelmingly dominant, with a direct path coefficient of 0.91 (Figure 7d). The direct path coefficients of the remaining factors on *LET* were very small and negative. The direct path coefficients of *T_a_*, *RH* and *WS* on *LET* were smaller than the sum of the indirect path coefficients. Consequently, the effects of these factors on *LET* were mainly reflected in the indirect effects, whereas the effects of *R_n_*, *VPD* and *SWC* on *LET* were primarily manifested through direct effects.

## 4. Discussion

### 4.1. Characteristics of Energy Partitioning

Agro-ecosystems are dynamic systems driven by solar radiation, where processes such as energy flow, material synthesis transfer and water–carbon cycle take place. These processes can vary significantly due to factors like crop types, growth periods, diurnal and seasonal fluctuations, climate conditions and anthropogenic disturbances to the underlying surface [42,43]. These variations can result in differences in the distribution of various energy components within the agro-ecosystem following the entry of net radiation [43].

The flux of *R_n_* is mainly influenced by regional solar radiation and surface vegetation albedo, which is stronger during the daytime, with the highest *R_n_* in the study area exceeding 700 W·m^−2^, which is higher than the values observed in agricultural fields in some areas in previous studies [44,45]. During the nighttime, when solar radiation is absent, *R_n_* tended to 0, often accompanied by sensible thermal advection [9,46], and *LET* was found to be greater than *R_n_* in the pineapple fields. In terms of the partition of energy expenditure, *H* and *LET* are the main forms of energy expenditure in pineapple fields, while *G* plays a lesser role. The sensible heat exchange is driven by the turbulent heat exchange coefficient and temperature gradients. A rougher underlying surface typically leads to greater temperature gradients and, consequently, higher *H* values [47]. Our study shows that *H* is the primary consumer of *R_n_* in pineapple fields, and similar results were observed in pineapple fields during the dry season in Venezuela [48], aligning with observations in various other ecosystems, including *Pinus tabuliformis* forests [49], plateau alpine meadow [50], semiarid shrublands [43], desert ecosystems [47] and *Nopalea cochenillifera* habitats [34]. It appears that these studies were in arid or semiarid ecosystems and that the water consumption (transpiration) intensity of these plants was relatively low.

Unlike our results, in the tropical environment of Brazil, the energy partitioning of pineapple fields is dominated by *LET* [22], which may be attributed to the increase in *LET* caused by sufficient irrigation. In southern subtropical China, the vast majority of pineapple crops are grown in rain-fed systems without irrigation facilities. In addition, many previous studies have reported that the energy partitioning of agro-ecosystems is dominated by latent heat transport, with *LET* in wheat and maize fields in the North China Plain accounting for over 83% of the available energy [23]. In rotationally cropped rice–wheat fields in the Yangtze River Basin of China, the *LET* in wheat fields accounted for 71% of the *R_n_*, whereas the *LET* in rice fields during the reproductive period was higher than the *R_n_* by 6% to 22% [51]. In the Loess Plateau of northwest China, *LET* accounts for 10% of *R_n_* in a rain-fed spring maize field [8]. This is mainly attributed to the fact that the study was conducted on C_3_ and C_4_ crops with strong canopy transpiration under water supply, which indicates that different vegetation cover on the underlying surface of the farmland played a significant role effect on energy partitioning. The canopy transpiration of these crops is the main component of farmland evapotranspiration, leading to higher *LET*.

Since the specific heat capacity of the soil is much greater than that of the air, the temperature change of soil in pineapple fields is later than that of air, so the time of *G* reaching the peak is lagging behind that of *R_n_*. Although *G* accounts for a very small proportion of *R_n_*, *G* has the phenomenon of absorbing energy during the daytime and releasing energy from the soil to the atmosphere on cloudy and at nighttime, and it acts as an “energy buffer” for the energy balance of the field [52]. In the study of energy balance in pineapple fields, *G* cannot be ignored.

### 4.2. Effects of Environmental Factors on LET

*LET* is of crucial significance in agricultural water management and evaluation of crop growth and water efficiency [9]. In the present study, the results of correlation and path analysis indicated that *R_n_* was the most significant environmental factor affecting the change of *LET*, followed by *VPD*. This finding aligns with similar conclusions drawn in various ecosystems [5,8,16,51,53], emphasizing the significance of *R_n_* in influencing *LET.* It is well known that the driving factors of *LET* are intricate, as its variation is influenced by a combination of biophysics and plant physiology. The path analysis shows that the primary influence of *R_n_* on *LET* in pineapple fields was mainly reflected in direct effects. The remaining environmental factors were mainly reflected in indirect effects on *LET* through the *R_n_* pathway, which indicated the key role of energy supply in energy partitioning in pineapple fields. Similar results were obtained in the study of factors affecting latent heat flux in rotated rice and wheat fields by Qiu et al. [53]. The correlation coefficient between *RH* and *LET* was found to be negative on sunny days, indicating that elevated *RH* inhibits *LET*. However, this inhibitory effect was not significant on cloudy days.

*VPD* characterizes the combined effect of *T_a_* and *RH* and serves as an important indicator to measure the degree of air dryness [54,55]. It is also another important factor in promoting *LET* in the pineapple field, and its impact is mainly indirect through *R_n_*. Several previous studies have documented the negative effects of high *VPD* on *LET* by inducing leaf stomatal closure [43,56,57], indicating the importance of bioregulation on transpiration and *LET* [58], but the negative effect of high *VPD* on *LET* under the pineapple CAM pathway did not appear in the present study. Surface *SWC* is the main source of *LET.* The effect of *SWC* on *LET* in pineapple fields passed the test of significance, but the correlation coefficient was relatively low, which may be related to other factors such as net radiation and air temperature being relatively low on cloudy and rainy days with high soil moisture, where changes in soil moisture have nonlinear impacts on *LET*. This is consistent with previous findings that there is a nonlinear response of energy partitioning to changes in *SWC*, with *LET* being less sensitive under higher *SWC* conditions [25,57,58,59], which differed from the findings from ecosystems with much drier environments [43,60].

Finally, this study was conducted over two years on a single season of pineapple fields. While our results are valuable and offer new insights, they should be complemented by further experiments to gain a more comprehensive understanding of the underlying processes governing energy fluxes in this CAM plant. Future research endeavors could involve distinguishing between the contributions of plant transpiration and soil evaporation to *LET* during the pineapple growth period, quantifying CO_2_ fluxes in pineapple field soils and the surrounding environment and examining the effects of adequate irrigation on energy partitioning in pineapple fields.

## 5. Conclusions

In this study, we conducted energy flux measurements in pineapple fields located in southern subtropical China, by means of the BREB system. The study focused on characterizing energy partitioning within these fields and identifying key factors influencing *LET*. *H*, *LET*, *G* and *R_n_* in pineapple fields showed similar temporal trends. During the daytime, the mean values of the Bowen ratio for pineapple fields were 2.09 and 1.41 for sunny and cloudy days and 0.74 and 0.46 for nighttime, respectively. *H* accounts for the majority of energy expenditure in pineapple fields, followed by *LET*, which accounts for 65.0% and 30.8% of *R_n_*, respectively. *G* accounts for a small portion of energy expenditure (4.2%). The energy input *R_n_* is the most dominant environmental factor affecting *LET* in pineapple fields, followed by *VPD*, with the effect of *T*, *RH*, *WS* and *SWC* on *LET* being relatively small. The path analysis indicates that the effect of *R_n_* on *LET* was mainly reflected in direct effects, while other factors were mainly reflected in indirect effects on *LET* through the *R_n_* pathway. Our research is informative for enhancing the impact of CAM plants on the surface energy balance and regional climate in the south subtropical climatic region, and the results offer valuable insights for decision-makers in agronomic cultivation, ecosystem management and climate change.

## Figures and Tables

**Figure 1 plants-13-00021-f001:**
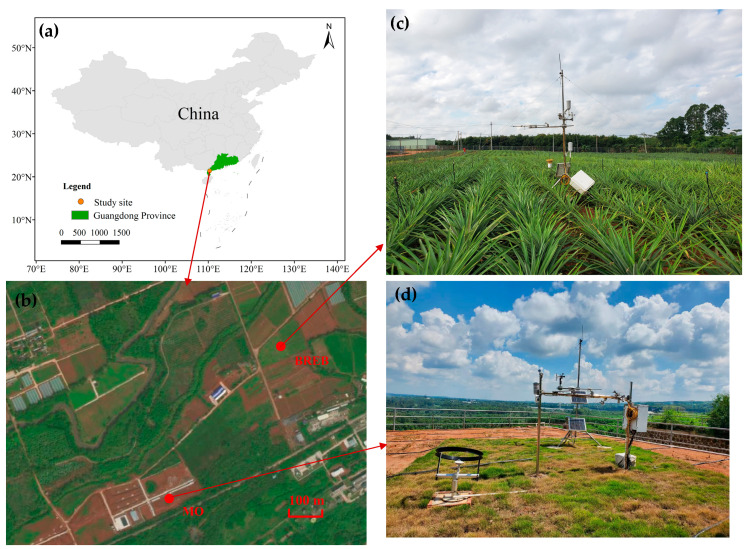
The layout and field photos of the experiment: (**a**) Study site; (**b**) Experimental station; (**c**) Bowen ratio energy balance system (BREB); (**d**) Meteorological observation station (MO).

**Figure 2 plants-13-00021-f002:**
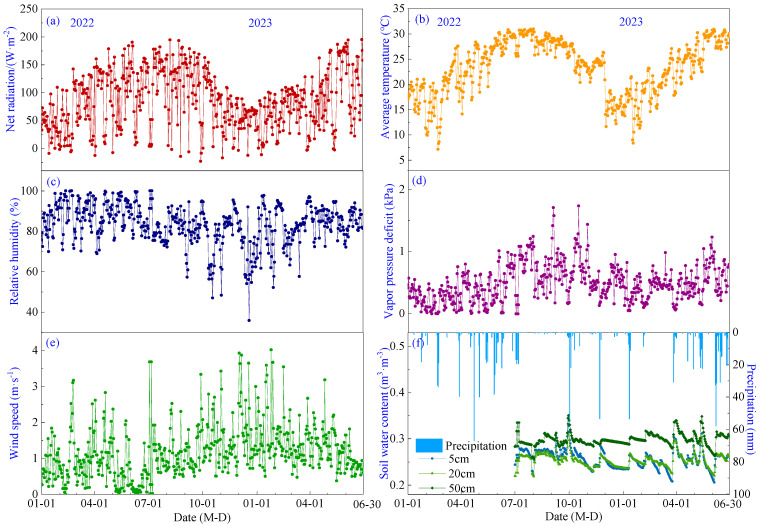
Seasonal variations of meteorological factors and soil water content during the growing period of pineapple: (**a**) Net radiation; (**b**) Air temperature; (**c**) Relative humidity; (**d**) Vapor pressure deficit; (**e**) Wind speed; (**f**) Precipitation and soil water content.

**Figure 3 plants-13-00021-f003:**
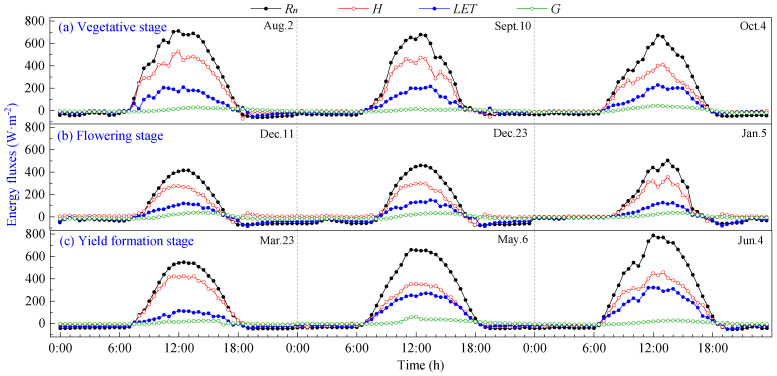
Daily variation curves of energy fluxes on sunny days of pineapple fields: (**a**) Vegetative stage; (**b**) Flowering stage; (**c**) Yield formation stage.

**Figure 4 plants-13-00021-f004:**
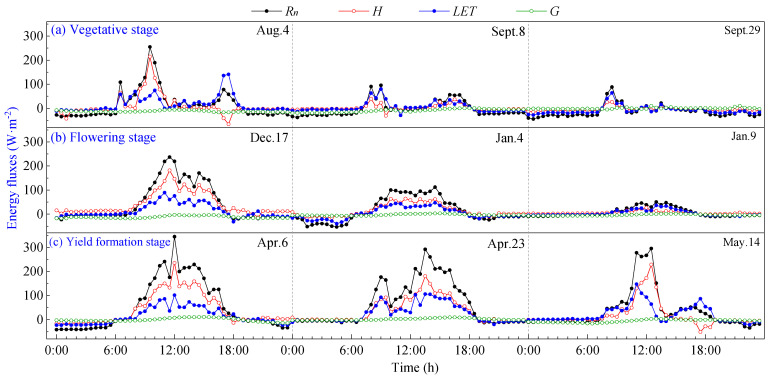
Daily variation curves of energy fluxes on cloudy days of pineapple fields: (**a**) Vegetative stage; (**b**) Flowering stage; (**c**) Yield formation stage.

**Figure 5 plants-13-00021-f005:**
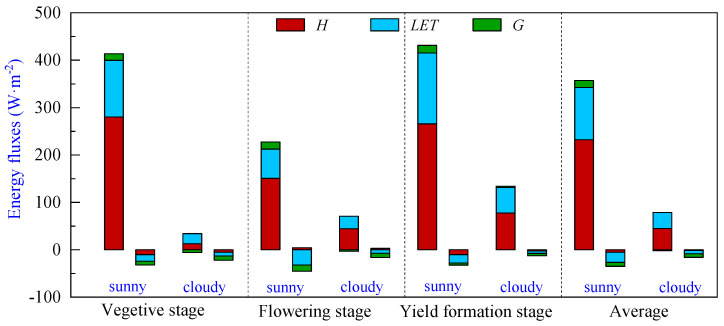
Energy partitioning of pineapple fields during different growth periods: sensible heat flux (*H*), latent heat flux (*LET*) and soil heat flux (*G*). Note: The left bar represents daytime (8:00–18:00), and the right bar represents nighttime (18:00–8:00) energy partitioning for each sunny or cloudy condition.

**Figure 6 plants-13-00021-f006:**
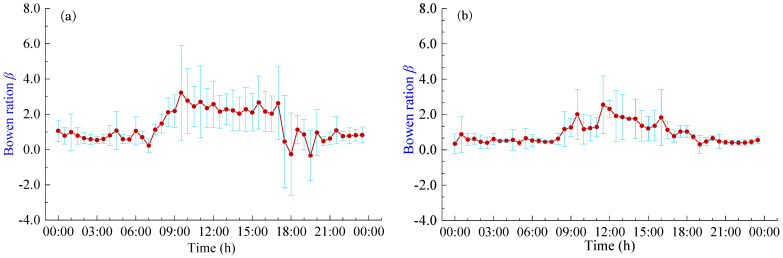
Daily variation of Bowen ratio *β* under different weather conditions: (**a**) Sunny days; (**b**) Cloudy days.

**Figure 7 plants-13-00021-f007:**
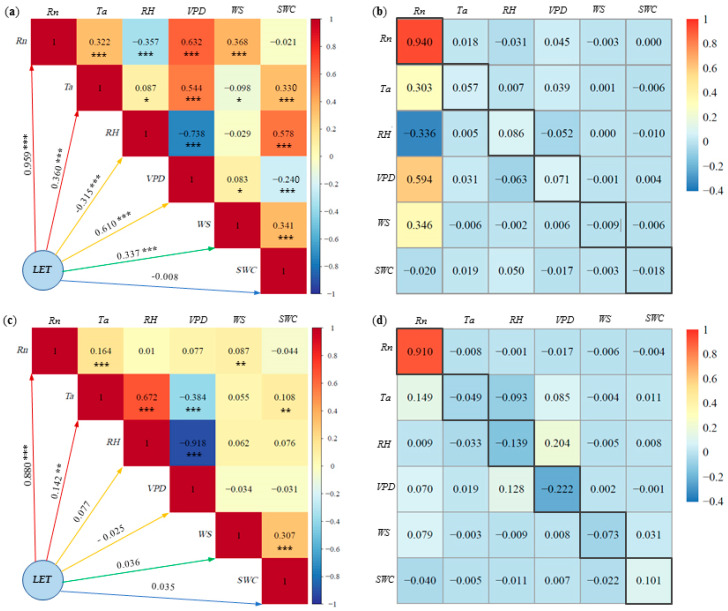
Latent heat fluxes (*LET*) and environmental factors of: (**a**) correlation analysis on sunny days, (**b**) path analysis on sunny days, (**c**) correlation analysis on cloudy days, (**d**) path analysis on cloudy days. * Significant at *p* < 0.05; ** significant at *p* < 0.01; *** significant at *p* < 0.001. Note: The number in the black frame is the direct path coefficient.

**Table 1 plants-13-00021-t001:** Conditions to be satisfied by the BREB method under nonadvective conditions for data to be reliable.

Available Energy	Vapor Pressure Gradient	Bowen Ratio	Heat Fluxes
*R_n_* − *G* > 0	Δ*e* > 0	*β* > −1	*LET* > 0 and *H* ≤ 0 for −1< *β* ≤ 0 or *H* > 0 for *β* > 0
	Δ*e* < 0	*β* < −1	*LET* < 0 and *H* > 0
*R_n_* − *G* > 0	Δ*e* > 0	*β* < −1	*LET* > 0 and *H* < 0
	Δ*e* < 0	*β* > −1	*LET* < 0 and *H* ≥ 0 for −1 < *β* ≤ 0 or *H* < 0 for *β* > 0

Source: Perez et al. [41].

**Table 2 plants-13-00021-t002:** Statistical values of energy partitioning of pineapple fields during different growth periods.

Growth Periods	Time Period	Sunny Days				Cloudy Days			
		*R_n_*	*H*	*LET*	*G*	*R_n_*	*H*	*LET*	*G*
Vegetative stage	8:00–18:00	413.44	280.35	119.47	13.62	28.96	12.57	21.41	−5.67
	18:00–8:00	−32.66	−10.29	−13.91	−8.11	−21.69	−5.13	−8.13	−8.78
Flowering stage	8:00–18:00	227.36	150.78	61.73	14.86	67.42	44.20	26.53	−3.31
	18:00–8:00	−41.18	4.34	−32.32	−13.20	−13.48	2.84	−7.40	−8.91
Yield formation stage	8:00–18:00	431.33	265.70	149.52	16.11	134.14	77.56	53.77	2.75
	18:00–8:00	−32.39	−10.33	−17.68	−4.38	−12.52	−2.52	−5.00	−5.20
Average	8:00–18:00	357.38	232.28	110.24	14.86	76.84	44.78	33.91	−2.07
	18:00–8:00	−35.41	−5.43	−21.30	−8.56	−15.90	−1.61	−6.84	−7.63

Note: *R_n_* is the net radiation; *H* is the sensible heat flux; *LET* is the latent heat flux; *G* is the soil heat flux.

## Data Availability

The data presented in this study are available upon request from the corresponding authors.

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
