# Peer review of "Energy Partitioning and Latent Heat Flux Driving Factors of the CAM Plant Pineapple (Ananas comosus (L.) Merril) Grown in the South Subtropical China"

_plants, 2023, doi:10.3390/plants13010021_

Round 1

Reviewer 1 Report

Comments and Suggestions for Authors

Figure 2: The figure presents the energy fluxes for three different stages. The vertical axis of the plots in the middle row (flowering stage) have a different range as compared to the other stages. Suggestion: make the three rows with the same values range in the vertical axis. That would ease visual comparison. Same comment for Figure 3.

Figure 4: The Figure presents four segments with four bars in each segment. The bars in each segment represent energy fluxes values in sunny (first and second blocks) and cloudy (third and fourth blocks). Having two blocks for each condition (sunny and cloudy) represent day and night, am I right? The text above the Figure refers to day and nighttime but the Figure and its caption should be self-explanatory. Perhaps a more clear caption is needed in this Figure.

Line 255: "...with a direct path coefficient of 0.96." Is it not 0.94?

Line 265: What is VWC?

Suggestion: Present the Methods just after the Introduction section. That would increase understanding of the Results for the reader who orderly reads the document from beginning to end.

Reviewer 2 Report

Comments and Suggestions for Authors

This study presents valuable results in evaluating the heat balance based on measured data. However, the authors' discussion based on the analysis results has fatal inconsistencies, as shown below.

The authors stated the following in section 3.2 (lines 318-322). As stated at the beginning of this explanation, VPD directly affects LET, while Rn affects LET via changes in VPD, via changes in T and RH. Therefore, the last part of this explanation that Rn directly affects LET is incorrect. Evidently, Rn affects LET indirectly via T, RH, and VPD.

"Rn can cause changes in T and RH, increase VPD, and enhance evapotranspiration rate. "This finding aligns with similar conclusions drawn in The primary influence of Rn on LET in pineapple fields was mainly reflected in direct effects. mainly reflected in direct effects."

Therefore, the conclusion of this study, ”The LET in pineapple fields was mainly affected by two key factors: Rn and VPD, with Rn having the most pronounced direct effect and VPD having an indirect effect on LET mainly through the Rn pathway” (lines 318-322) is incorrect. Since Rn affects LET via VPD, it is obvious that Rn is indirect and VPD is direct.

Round 2

Reviewer 2 Report

Comments and Suggestions for Authors

In my previous point, I stated “Evidently, Rn affects LET indirectly via T, RH, and VPD.” However, the authors maintained that “Rn directly supplies the energy required for LET (or evapotranspiration).” Therefore, I cannot change my previous assessment because the authors ignored my point.

Author Response

Thank you very much for your comments. We have carefully considered your point of view and recognize it to some extent. Our existing results and conclusions are obtained based on the path analysis and have been described and rigorously argued in the manuscript.

It is well known that the change of LET is related to bio-atmospheric physics and plant physiology. LET (or evapotranspiration) is the energy absorbed or released by water when its physical state changes. The conditions for controlling LET are: (1) vegetation and water supply; (2) energy conditions: it depends on the amount of energy obtained by water molecules on the evaporation surface, including net radiation, temperature, etc. (3) dynamic conditions: that is, water vapor transport conditions in the atmosphere, including VPD, wind speed, etc. The conclusions based on path analysis can also explain and verify the above phenomenon to some extent. Thank you again for your constructive comments and suggestions on our manuscript.